# Cytogenetic Effects in Children Exposed to Air Pollutants: A Systematic Review and Meta-Analysis

**DOI:** 10.3390/ijerph19116736

**Published:** 2022-05-31

**Authors:** Mattia Acito, Cristina Fatigoni, Milena Villarini, Massimo Moretti

**Affiliations:** Unit of Public Health, Department of Pharmaceutical Sciences, University of Perugia, Via del Giochetto, 06122 Perugia, Italy; mattia.acito@studenti.unipg.it (M.A.); cristina.fatigoni@unipg.it (C.F.); milena.villarini@unipg.it (M.V.)

**Keywords:** micronucleus assay, children, air pollutants, air pollution, human biomonitoring

## Abstract

The aim of this systematic review and meta-analysis was to assess the association between exposure to ambient air pollutants and micronuclei (MN) frequency in children. This work was performed according to the Cochrane Collaboration and the PRISMA guidelines and recommendations. Articles published before November 2021 were identified by an advanced search on PubMed/MEDLINE, Scopus and Web of Science databases. A critical appraisal using a specific tool was conducted to assess the quality of each included study. All analyses were carried out by using the Review Manager (RevMan) 5.4 software (The Cochrane Collaboration, London, UK). One hundred and forty-five references were firstly identified, and, at the end of selection process, 13 studies met the inclusion criteria. Six studies carried out a direct evaluation through the use of air samplers, whereas the other ones accessed environmental databases (n = 2) or used other tools (n = 3). In two cases, exposure was not directly investigated, with children sampled in two different areas with well-known different levels of pollution. The overall effect size (ES) was 1.57 ((95% CI = 1.39; 1.78), *p*-value < 0.00001) (total evaluated subjects: 4162), which highlighted a statistically significant association between outdoor air pollution and MN frequency in children. As a high MN frequency has been associated with a number of pathological states and a higher risk of developing chronic degenerative diseases, our results should be taken into consideration by policy makers to design and implement interventions aimed at reducing the introduction of pollutants in the atmosphere as well as at minimizing the exposure extent, particularly in children.

## 1. Introduction

Air pollution could be defined as the presence in the atmosphere of one or more substances at a concentration or for a duration above their natural levels, with the potential to produce adverse effects on health and/or the environment [1]. Outdoor air pollution is a complex mixture of thousands of different compounds including gaseous pollutants, such as carbon monoxide (CO), nitrogen dioxide (NO_2_), nitrogen oxides (NO_X_), sulfur dioxide (SO_2_), ozone (O_3_), volatile organic compounds (VOCs), such as benzene, heavy metals, such as lead (Pb), and particulate matter (PM), with air pollutants typically classified as primary or secondary. Primary pollutants are chemicals, such as CO, CO_2_, NO_X_, SO_2_, and PM released directly into the atmosphere mainly produced by combustion of fossil fuels (e.g., motorized road traffic, power generation, industrial activities, and residential heating). In contrast, secondary pollutants, such as PM formed from secondary organic aerosols and O_3_, are formed by chemical reactions occurring in the atmosphere between different primary pollutants or between primary pollutants and atmospheric gases. Ground-level O_3_ is a typical secondary pollutant formed in the troposphere through a complex series of reactions involving the action of sunlight on NO_2_ and VOCs [2]. Among these pollutants, PM, primary or secondary in origin, consists of particles commonly gathered into three size groups according to its nominal median aerodynamic diameter (AD): coarse particles with AD 2.5–10 µm (PM_10_), fine particles with ADs less than 2.5 µm (PM_2.5_), and ultrafine particles with ADs less than 0.1 µm [3]. The complex mixtures of air pollutants (i.e., gaseous and particulate-bound pollutants) may be extremely heterogeneous in their composition, depending on human activities and meteorological conditions in a particular geographical area [4].

Ambient air pollution is a major public health problem worldwide, affecting people resident in most of urban areas. According to the World Health Organization (WHO) Ambient Air Pollution Database (covering 4300 settlements in 108 countries) [5], more than 80% of people living in urban areas that monitor air pollution are exposed to air quality levels that exceed the WHO limits [6]. In the European Union, 97% of the urban population is exposed to levels of fine PM above the latest guideline levels set by the WHO [7], and in the 27 member states of the European Union, 307,000 premature deaths are possibly associated to exposure to fine PM [7]. Even though populations in low-income cities are the most impacted (with people deriving energy from natural, inexpensive sources without having adequate technologies to mitigate potential air pollutants) [8], according to the WHO, 56% of cities in high-income countries with more than 100,000 inhabitants do not meet WHO air quality guidelines [6].

In Europe, the most recent directive relating to air quality is the Directive 2008/50/EC or Air Quality Directive (AQD), which entered into force in June 2008 [9]. The AQD sets limit values for the ambient (outdoor) concentrations to be achieved for several pollutants which are harmful to human health: CO, NO_2_, NO_X_, SO_2_, Pb, benzene, O_3_, and PM. The AQD is currently among the strictest acts of legislation worldwide concerning PM_10_ and PM_2.5_ air pollution [10] by establishing for PM_2.5_ a target value of 25 µg/m^3^, as an annual average. Target values for specific heavy metals, such as arsenic, cadmium, and nickel (only monitoring requirements are specified for mercury), and polycyclic aromatic hydrocarbons (PAHs) in ambient air are reported in the Directive 2004/107/EC [11]; for PAHs, the target is defined in terms of concentration of benzo(a)pyrene (BaP), which is used as a marker substance for PAHs generally.

In the United States, the Clean Air Act (CAA) is major legislation passed to control air pollution. The CAA was last amended in 1990 with the federal Environmental Protection Agency (EPA) having the responsibility for establishing standards and enforcing the Act [12]. The US EPA has set National Ambient Air Quality Standards for six principal pollutants, defined as “criteria” air pollutants, considered harmful to public health; they are PM, ground-level O_3_, CO, sulfur oxides (SO_X_), NO_X_, and Pb. Where a criteria pollutant is actually a group of pollutants (e.g., NO_X_), the standards are set for key or indicator pollutants within the group (e.g., NO_2_).

Current air pollution frequently found in urban areas is thus a dynamic and complex mixture of pollutants of both anthropogenic (e.g., traffic, residential heating, industry, etc.) and natural origin. It is generally accepted that exposure to these (toxic/genotoxic) agents in the atmospheric compartment poses serious implications for human health [13]. Short-term exposure to ambient air pollution has been associated to exacerbated asthma responsible for increased hospital admissions [14], whilst long-term exposure to airborne pollutants has been reported to be associated with a higher incidence of cardiovascular and respiratory diseases [15,16], birth defects [17], and neuro-degenerative disorders [18]. The WHO estimated that outdoor air pollution may have caused 3.7 million premature deaths worldwide in 2012 [19]. According to the WHO, 14% of deaths were caused by obstructive pulmonary disease or chronic respiratory infections, 6% by lung cancer, and approximately 80% were caused by ischemic heart disease and stroke [19]. PM is among the most studied air pollutants of health concern [20] and, in agreement with the global burden of disease reported for 2010 [21], in a review on global mortality, it has been reported that outdoor air pollution, mostly PM_2.5_, leads to 3.3 million premature deaths per year worldwide [22]. In 2013, the International Agency for Research on Cancer (IARC) evaluated the carcinogenic risk to humans of outdoor air pollution. The position of the IARC was firstly released to the public in a short communication reporting that the Working Group “classified outdoor pollution and particulate matter from outdoor air pollution as carcinogenic to humans (Group 1)” [23]; the results of this assessment were then published as Volume 109 of the IARC Monographs [24].

Several studies have recently highlighted the evidence of an association between socio-economic status and other deprivation indices with exposure to high levels of outdoor air pollution [25,26,27]. On one hand, exposure to air pollutants is strongly driven by environmental inequalities [28]; on the other, it affects vulnerable groups—such as children and the elderly—more than others [29,30].

In general, epidemiological evidence has shown that children are more sensitive than adults to genotoxic agents and that genetic damage appearing at younger ages may affect the lifetime risk of adverse health outcomes (e.g., cancer) [31]. In particular, children are considered to be a high-risk group in terms of the health effects of air pollution [32,33] because of their different and unique pathways of exposure, their dynamic developmental physiology and their longer life expectancy [34]. Some studies have suggested that early exposure during childhood can play an important role in the development of chronic diseases in adulthood [35,36,37,38,39]. The higher susceptibility of children, with respect to adults, to the noxious effects of air pollution might depend on smaller airways, immature detoxification and metabolic systems, as well as frequent exposure to outdoor air of children [40].

As long-term health adverse effects, such as chronic obstructive pulmonary disease, cardiovascular disease or cancer, of moderate or low air pollution levels might not be clearly highlighted by classic epidemiology, especially in small-scale studies [41], there is a growing number of molecular epidemiology studies using genotoxicity biological markers to study the effects of exposure to environmental pollutants [42,43,44]. Biomonitoring of genotoxic hazards has been reported in several studies by the use of different genotoxicity endpoints, such as analysis of primary DNA damage (by the comet assay), or cytogenetic effects, such as micronuclei (MN), and sister chromatid exchanges [45].

Among genotoxicity endpoints, MN is one of the most commonly used biomarkers in studies evaluating environmental or occupational risks associated with exposure to potential genotoxins [46]. MN testing, because of its ability to detect both clastogenic (e.g., chromosome breakage) and aneugenic (e.g., spindle disruption) effects, is considered to be a biomarker of early biological effect [47,48]. MN appears in the cytoplasm of interphasic cells as small additional nuclei that are smaller than the main nucleus. MN typically generate during the anaphase from acentric chromosome fragments (chromosome breakage produced by clastogen agents) or whole chromosomes (chromosome malsegregation caused by aneugen agents). Acentric or whole chromosomes are left behind during mitotic cellular division and, consequently, are excluded from both of the daughter nuclei [49,50,51,52].

The lymphocyte cytokinesis–block micronucleus (L-CBMN) assay is one of the most frequently used and, to date, the best validated method for biological effect monitoring in subjects with residential/occupational exposure to genotoxic xenobiotics [53,54,55,56]. The frequency of MN in circulating lymphocytes is recognized to be a predictor of cancer risk in human populations [57,58,59]. The assessment of MN in uncultured, exfoliated epithelial cells from oral mucosa (B-MN assay) has later provided a complementary method for cytogenetic analysis in an easily accessible tissue without cell culture requirement [60].

In this study, we identified and analyzed the studies published reporting the use of MN assay—either using the L-CBMN or the B-MN assay—as a biomarker of genotoxic risk in children exposed to air pollutants with the aim of performing a meta-analysis of data and providing a meta-estimate of the genotoxic effect of exposure.

## 2. Materials and Methods

This systematic review and meta-analysis was carried out following the Cochrane Collaboration [61] and Meta-analysis of Observational Studies in Epidemiology (MOOSE) guidelines [62], and results were reported according to the Preferred Reporting Items for Systematic Reviews and Meta-Analyses (PRISMA) statement [63,64,65].

### 2.1. Search Strategy and Data Sources

An extensive literature search was carried out in November 2021 through the use of the PubMed/MEDLINE (National Library of Medicine, National Institutes of Health, Bethesda, MD, USA—http://www.ncbi.nlm.nih.gov/PubMed (accessed on 30 November 2021)), Scopus^®^ (Elsevier, Amsterdam, Netherlands—https://www.scopus.com (accessed on 30 November 2021)) and the Web of Science (Clarivate Analytics, London, UK—https://www.webofknowledge.com (accessed on 30 November 2021)) databases. The literature search was conducted using a pre-determined combination of keywords, considering the following features: micronuclei, children and air pollution. Both Medical Subject Headings (MeSH) and text words were used in this step. Keywords were combined using Boolean operators AND/OR. The strategy was first developed in PubMed/MEDLINE and then adapted for the other databases (Appendix A). The reference lists of related articles were also reviewed for additional relevant studies.

### 2.2. Inclusion and Exclusion Criteria

Appendix A illustrates a description of inclusion and exclusion criteria according to the Population, Exposure, Outcomes and Study design (PEOS) [66], implemented, according to the Cochrane Collaboration [67], with time and language filters.

Papers were considered eligible if the studies met the following inclusion criteria: (P) human biomonitoring of healthy children in young populations including age range 5–12; (E) populations exposed to at least two different levels of environmental pollutants; (O) and (S) evaluation of micronuclei in donors’ cells within cross-sectional studies. Finally, the provision of group mean values with standard deviation or sufficient data (e.g., plots) to calculate them and the use of the English language were considered among inclusion criteria as well.

### 2.3. Study Selection and Data Extraction

The title and abstract of the retrieved records were blindly screened by two different researchers (M.A. and C.F.) in order to collect potentially relevant articles. Full text was obtained only for selected reports that met inclusion/exclusion criteria. Data were independently extracted from included studies by two different authors (M.A. and C.F.) and transferred onto a Microsoft Word^®^ (Redmond, WA, USA) document. Pre-arranged tables were used to systematically record qualitative and quantitative features extracted from the included reports. In case of incomplete data, corresponding authors were contacted by e-mail. For line graphs, the data were extracted from the graphics using WebPlot Digitizer 4.51 software (https://automeris.io/WebPlotDigitizer/index.html (accessed on 19 January 2022)), Automeris LLC, Pacifica, CA, USA [68].

Any disagreement in both record screening and data extraction was solved through discussion between the two researchers and, in case of disagreement persistence, a third author (M.M.) was consulted.

### 2.4. Critical Appraisal

The critical appraisal of the included reports was independently performed by two researchers (M.A. and C.F.) using a specific tool developed for the quality assessment of studies using MN frequency as a biomarker of chromosomal damage [69]. In this case, the quality score sheet was adapted to studies investigating children populations by removing “alcohol intake” item and by remodeling “smoking status matching” as “passive smoking status matching”. The quality score (QS), which can range from 8 (the poorest quality) to 24 (the highest quality), was calculated for each study and tabulated with the other characteristics of the included articles.

### 2.5. Meta-Analysis

Reported MN frequencies could be affected by differences in the adopted scoring criteria [70,71]. Consequently, as a measure of the effect size, for each study included in the meta-analysis, the Ratio of Means (RoM) was computed, with RoM being defined as the mean value in the exposed group divided by the mean value in the control group [72]. The main advantage of RoM is the possibility to compare and pool studies with outcomes that are expressed in different units. Moreover, RoM represents a measure of the effect size that is relatively independent from the absolute value assumed by the means and, consequently, it is poorly affected by inter-laboratory variability [72].

This meta-analysis was conducted applying the generic inverse variance method, calculating the natural logarithm of each study RoM and its standard error (SE) [72].

Analyses were performed using Review Manager (RevMan) 5.4, The Cochrane Collaboration meta-analysis calculation software (London, UK), freely available at: https://training.cochrane.org (accessed on 8 February 2022). Heterogeneity was estimated through χ^2^ and I^2^ tests. The I^2^ value represents the percentage of the total variation across studies due to heterogeneity within a group and across a group; it takes values from 0% to 100%, with the value of 0% indicating no observed heterogeneity. According to the Cochrane Collaboration [67], heterogeneity was identified as follows: 0% to 40%: might not be important; 30% to 60%: may represent moderate heterogeneity; 50% to 90%: may represent substantial heterogeneity; 75% to 100%: considerable heterogeneity. Fixed or random effects were used in this study according to the heterogeneity. As a fixed-effects model should be used only if it reasonable to assume that all studies share the same common effect, it is commonly accepted to use a random effect model in case of high heterogeneity (I^2^ ≥ 50%). Potential publication bias was assessed visually by evaluating Begg’s funnel plots asymmetry.

### 2.6. Sensitivity Analysis

In order to assess the robustness of results, we also performed sensitivity analysis only considering studies with a high/medium QS (QS ≥ 13) and studies involving a number of subjects ≥ 20 in both exposed and control groups.

## 3. Results

### 3.1. Literature Search

Our electronic searches yielded 145 references, with 87 remaining after duplicate (n = 58) removal. We preliminarily screened the titles and abstracts of these records and, after excluding articles written in languages other than English (n = 14), we identified 73 as potentially eligible and obtained the full text for 72 reports. After that, a total of 59 were excluded for the following reasons: unrelated topic (n = 20), in vitro/plant/animal models (n = 9), reviews (n = 12), non-primary data, protocols or conference proceedings (n = 10), lack of two levels of exposure to be compared (n = 4), results expressed as median and interquartile range (n = 3), and results lacking standard deviation or standard error of the mean (n = 1). At the end of the screening procedure, 13 articles were included in the systematic review and meta-analysis (Figure 1).

### 3.2. Characteristics of the Included Studies

Characteristics of the included studies are reported in Table 1 and Table 2 and Appendix A. Five studies were carried out in Europe [73,74,75,76,77] and South America [78,79,80,81,82], and three studies were conducted in Asia [83,84,85]. The first paper was published in 2000 [85], whereas the most recent one was released online in December 2020 [83].

Exposure was assessed by different methodologies. Six studies carried out a direct evaluation through the use of air samplers, whereas the other ones accessed environmental databases (n = 2) or used other tools (n = 3). In two cases, exposure was not directly investigated, with children sampled in two different areas with well-known different levels of pollution [83,85].

In 11 out of 13 studies, populations exposed to at least two different levels of environmental pollution were geo-spatially separated, whereas in one study, the comparison was based on a temporal separation. In one study, exposure and MN frequency assessment was structured on both spatial and temporal level. In case of investigation of more than two areas within a study, the most and the less polluted areas were compared in our analysis.

Nine studies exclusively investigated MN frequency in buccal mucosa cells, two studies on blood cells and two of them in both cell types.

As regards the quality assessment, the score ranged between 11 [78,80] and 20 [84] (median value: 14). Overall, none of the included studies obtained the maximal score (i.e., 24); three investigations had a low QS (range 8–12) [78,79,80], two studies had a high QS (range 19–24) [74,84], whereas the rest achieved medium QS (range 13–18).

When considering the number of exposed and control subjects, seven studies recruited more than 50 subjects per group, whereas the sample size was lower than 20 in both groups in none of the studies. The highest number of recruited subjects (n = 1046) was found in [74]. In most of the included studies, subjects in the exposed and control groups were—at least partly—age- and gender-matched. However, perfect matching was found only in those studies in which exposure comparison and MN frequency were assessed in two distinct periods in the same population [74,84]. Passive smoking status and nutritional intake were unmatched or not reported in 8 and 10 studies out of 13, respectively, making them the main factors contributing to a low/medium QS.

In all the included studies, the number of cells scored per subject was equal to 1000 or 2000. No study scored less than 1000 or more than 2000 cells per subject.

### 3.3. Results of Meta-Analysis

Considering all the included studies, and using the random effect model, the pooled ES was 1.57 (95% CI = 1.39; 1.78) (Figure 2a), based on 4,162 participants, and heterogeneity was also calculated (Chi^2^ = 213.48, df = 16, I^2^ = 93 %, *p* value < 0.00001). A potential publication bias was found by the visual assessment of the funnel plot (Figure 2b).

### 3.4. Sensitivity Analysis

Sensitivity analysis was conducted in order to confirm the robustness of our results. Particularly, we first removed the studies with low QS (QS < 13) and subsequently those that recruited less than 20 subjects per group: the excluded studies were the same in both cases. We obtained similar results compared with the main analysis (Figure 3): the pooled ES was 1.59 (95% CI = 1.40; 1.82). Heterogeneity (Chi^2^ = 200.66, df = 11, I^2^ = 95%, *p* value < 0.00001).

By analyzing the five studies reporting the use of the L-CBMN assay, we estimated an overall effect size of 1.34 (95% CI: 0.98; 1.84) (Figure 4). Likewise, by considering the studies reporting the use of the B-MN assay in exfoliated cells, we estimated an overall effect size of 1.64 (95% CI: 1.42; 1.89) (Figure 5).

## 4. Discussion

This extensive systematic review and meta-analysis, conducted by searching three different databases (i.e., PubMed/MEDLINE, Scopus, and Web of Science), assessed the association between exposure to air pollutants and MN frequency in children. Our meta-analysis of 13 studies found a statistically significant increase (+57%) of MN frequency in populations exposed to air pollutants when compared with control groups. In most of the studies, populations were geo-spatially separated, whereas in a few cases, the same population was sampled twice in two differently polluted periods of the year. Results were confirmed also after performing a sensitivity analysis, where we excluded low-quality studies as well as studies recruiting less than 20 subjects per group. It should be noted that heterogeneity is high in both main and sensitivity analysis (I^2^ = 93% and 95%, respectively).

The great majority of studies included in this meta-analysis have been performed by MN assay on buccal cells. Obtaining venous blood from the median cubital or antebrachial veins is usually carried out in adults; anyway, this sampling procedure is certainly more cumbersome in children and sometimes poorly accepted by parents in the absence of risk perception. The assessment of MN in exfoliated epithelial cells from oral mucosa has thus provided a complementary method for cytogenetic analyses in an easily accessible tissue without cell culture requirement [60]. Nowadays, the human buccal micronucleus assay is one of the most widely used techniques to measure genetic damage in human population studies [60,70,86].

Overall, our results confirm the use of MN as an important and robust biomarker to monitor the genotoxic effects of chemical/toxic agents and/or their metabolites in children populations, particularly the B-MN assay. These results are also in line with a number of in vitro studies showing clastogenic/aneugenic effects of ambient air pollutants (e.g., PM, PHAs, etc.) in various experimental models and cell lines [87,88,89]. 

### Limitations and Strengths

Despite the statistically significant meta-estimate obtained in our work, the present systematic review and meta-analysis is affected by some limitations, which are mostly related to meta-analysis research in general. In this approach, we only selected studies published in English, so reporting bias (i.e., language bias) cannot be excluded. Publication bias was also investigated visually through the evaluation of funnel plot asymmetry. Particularly, there is a clue of missing studies in the middle/bottom of the plot, especially in the area of non-significance. As results from small studies usually scatter widely at the middle/bottom of the graph, we assume that small studies that report non-significant conclusion went unpublished and are underrepresented in the meta-analysis, making publication bias plausible. Furthermore, our analysis did not directly address some design elements, such as ambient air contaminants concentration, as not all of the considered studies reported adequate information (e.g., data on environmental or biological monitoring). There was also substantial heterogeneity among the studies considered as they were performed by different research teams in different places and settings with different populations. Our meta-analysis showed a high heterogeneity (*p* value for χ^2^ < 0.00001; I^2^ = 93%). The I^2^ value (directly related to τ^2^) found in our meta-analysis indicates that 93% of the total variability among effect sizes is not caused by sampling error but by true heterogeneity among studies [90]. This heterogeneity probably arises from differences in participant characteristics (e.g., age, gender, passive smoking, etc.) and exposure characteristics, as complex mixtures of air pollutants (i.e., gaseous and particulate-bound pollutants) may be extremely heterogeneous in their composition, depending on human activities and meteorological conditions in a particular geographical area [4].

The main strength of this systematic review and meta-analysis resides in its comprehensive consideration of the scientific evidence published so far on the main medical-scientific databases. Furthermore, the pooled meta-estimate was significantly large, compared on the sample size of single original studies, and it was based on 4162 children. Moreover, MN frequency measured in the L-CBMN or B-MN assay, even if occurring at different frequency, was shown to be highly correlated and hence to have a similar ability to detect effects of exposure to genotoxic agents in children. 

## 5. Conclusions

In summary, the results of this systematic review and meta-analysis show that exposure to a polluted air environment is statistically associated with a higher frequency of MN in children. Furthermore, our results confirm the sensitivity of both L-CBMN and B-MN assay in detecting cytogenetic effects induced by airborne pollutants. In particular, the B-MN assay is a simple, cost-effective, and non-invasive test which could be easily used for monitoring air pollution biological effects in children [91]. Moreover, important confounding factors such as passive smoking and nutritional intake were unmatched or not reported in most of the considered studies. To better define the role of ambient air pollution in induced chromosomal damage in children, this kind of bias should be taken into consideration in future research.

In conclusion, as a high MN frequency has been associated with a number of pathological states and a higher risk of developing chronic degenerative diseases, our results should be taken into consideration by policy makers to design and implement interventions aimed at reducing the introduction of pollutants in the atmosphere as well as at minimizing the exposure extent, particularly in children.

## Figures and Tables

**Figure 1 ijerph-19-06736-f001:**
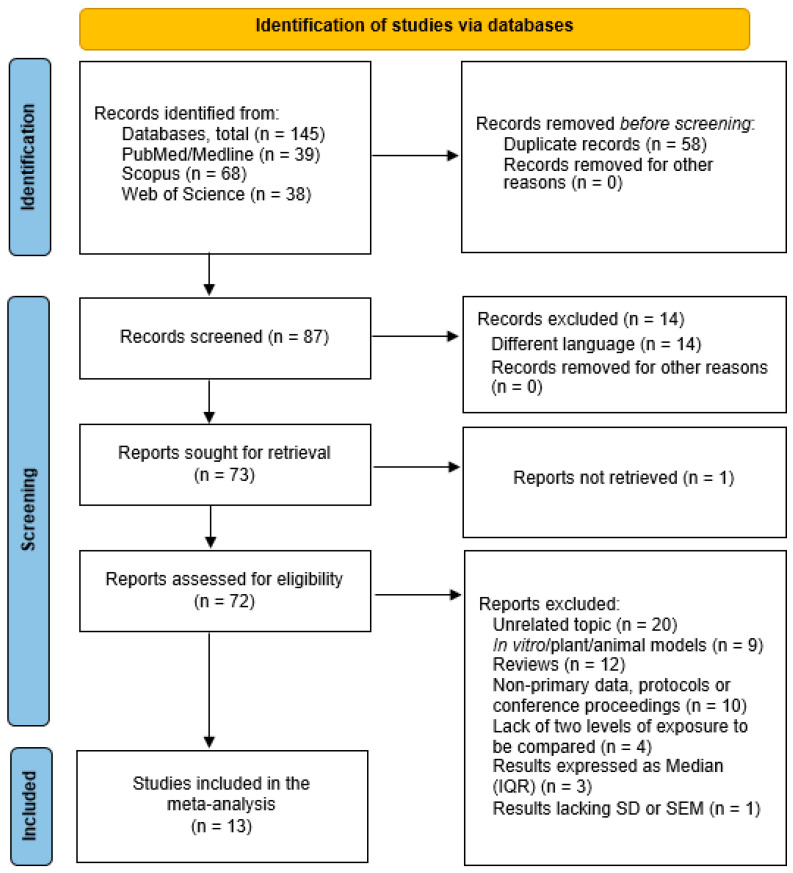
Flow diagram of the selection process according to the PRISMA 2020 statement [65].

**Figure 2 ijerph-19-06736-f002:**
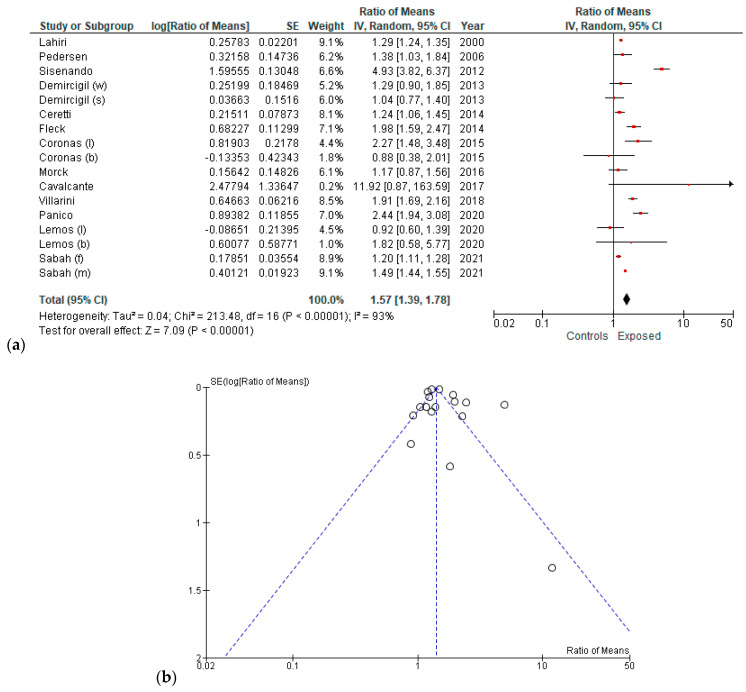
(**a**) Forest plot and (**b**) Funnel plot of the meta-analysis comparing exposure to ambient air pollution (lower vs. higher) and frequency of MN (random effect model). (w, s) winter and summer, respectively; (l, b) L-CBMN and B-MN assay, respectively; (f, m) females and males, respectively.

**Figure 3 ijerph-19-06736-f003:**
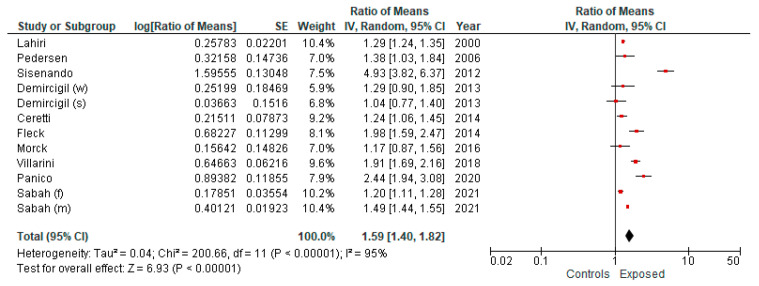
Forest plot of subgroup meta-analysis comparing exposure to ambient air pollution (lower vs. higher) and frequency of MN (random effect model) limited to studies with a QS equal or higher than 13 or to studies with a sample size comprising at least 20 or more subjects per group: the excluded studies were the same in both cases. (w, s) winter and summer, respectively; (f, m) females and males, respectively.

**Figure 4 ijerph-19-06736-f004:**
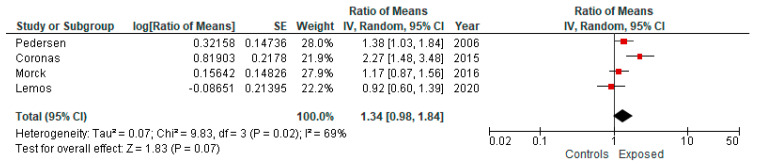
Forest plot of subgroup meta-analysis comparing studies reporting the use of the L-CBMN assay in comparing exposure to ambient air pollution (lower vs. higher) and frequency of MN (random effect model).

**Figure 5 ijerph-19-06736-f005:**
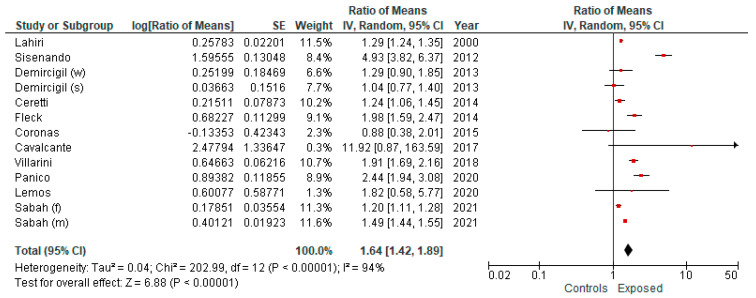
Forest plot of subgroup meta-analysis comparing studies reporting the use of the B-MN assay in comparing exposure to ambient air pollution (lower vs. higher) and frequency of MN (random effect model).

**Table 1 ijerph-19-06736-t001:** Main qualitative characteristics of included studies, reported in chronological order.

Author/s, Year [Ref.]	Country	Population Characteristics	Tool for Exposure Measurement	Funds ^[a]^	CoI ^[a,b]^
Sabah, 2021 [83]	Iraq	Schoolchildren living in close proximity to—or distant from—the Iraqi oil fields.	Exposure was not directly investigated. Children were recruited in an area in proximity or distant from an oil field.	Yes	n.d.
Lemos et al., 2020 [78]	Brazil	Children at two public schools in two different sites of the city of Triunfo.	High volume air samplers (AVG 1200/CCV Thermo Environmental Instruments) installed at the sampling sites.	Yes	No
Panico et al., 2020 [73]	Italy	Children at public schools in two different areas (more or less urbanized) of Southern Italy (Apulia Region).	High-volume air sampler equipped with multistage cascade impactor (AirFlow PM10-HVS sampler, AMS Analitica) installed at the sampling sites.	Yes	No
Villarini et al., 2018 [74]	Italy	Children at public schools in 5 Italian cities (Brescia, Turin, Pisa, Perugia and Lecce), sampled in winter and late spring.	Questionnaire.	Yes	No
de Carvalho Cavalcante et al., 2017 [79]	Brazil	Children attending two different schools in the city of Dourados: School A (high traffic area); School B (low traffic area).	The following formula was used to calculate the vehicular traffic intensity: VTI = Nv/T where VTI = vehicular traffic intensity; Nv = number of vehicles on a certain road, and T = duration of vehicle scoring (1 h).	Yes	No
Coronas et al., 2016 [80]	Brazil	Children residing in the surrounding wood treatment plant (city of Triunfo).	High volume air samplers (AVG 1200/CCV Thermo Environmental Instruments) installed at the sampling sites.	Yes	n.d.
Mørck et al., 2016 [75]	Denmark	Danish schoolchildren from the DEMOCOPHES population living in a urban (Gentofte) or rural (Viby Sjælland) area.	Questionnaire (traffic exposure). Air pollution levels were calculated using AirGIS system developed at Aarhus University.	Yes	n.d.
da Silveira Fleck et al., 2014 [81]	Brazil	Students at public schools in two areas of Porto Alegre (Protásio Alves Avenue, high population density; Juca Batista Avenue, low population density).	Passive sampling.	Yes	No
Ceretti et al., 2014 [76]	Italy	Healthy children living in different areas of the city of Brescia.	Questionnaire (traffic data). Environmental data were retrieved from the freely available ARPA (Regional Agency for Environmental Protection) database.	Yes	No
Demircigil et al., 2013 [84]	Turkey	Children attending two schools in the city of Eskişehir, sampled in summer and winter.	Passive samplers.	Yes	n.d.
Sisenando et al., 2012 [82]	Brazil	Schoolchildren living in two Brazilian areas: Tangará da Serra (industrial area) and Chapada dos Guimarães (rural area).	Data were obtained from CATT-BRAMS (Coupled Aerosol and Tracer Transport model of the Brazilian Regional Atmospheric Modeling System).	Yes	No
Pedersen et al., 2006 [77]	Czechia	Children living in two areas of Czech Republic (Teplice—mining area and Prachatice—less polluted area).	Samplings with a handheld condensation particle counter (TSI, model 3007) and a photometer (TSI, Dusttrack model 8520) equipped with a 2.5 μm impactor were performed.	n.d.	n.d.
Lahiri et al., 2000 [85]	India	Schoolchildren living in two areas of India (city of Calcutta and rural West Bengal).	Exposure was not directly investigated. Children were recruited in a highly urbanized area (Calcutta) and in a rural area (rural West Bengal).	Yes	n.d.

^[a]^ n.d., not declared. ^[b]^ CoI, conflict of interest.

**Table 2 ijerph-19-06736-t002:** Main quantitative characteristics of included studies, reported in chronological order.

Author/s, Year [Ref.]	Cell Type ^[a]^	Sample Size	MN Exposed	MN Controls	Fold-δ	*p* Value ^[b]^	QS/27 ^[c]^
Sabah, 2021 [83]	BMC	E: 100 NE: 100	M: 25.81 ± 2.89 F: 21.47 ± 4.04	M: 17.28 ± 1.94 F: 17.96 ± 1.38	1.49 1.19	*p* = 0.048 *p* = 0.05	15
Lemos et al., 2020 [78]	PBL	E: 28 NE: 8	1.66 ± 0.17	1.81 ± 0.34	0.92	*p* > 0.05	11
	BMC	E: 29 NE: 21	0.31 ± 0.13	0.17 ± 0.07	1.82	*p* > 0.05	11
Panico et al., 2020 [73]	BMC	E: 206 NE: 256	0.66 ± 0.61	0.27 ± 0.43	2.44	*p* < 0.001	17
Villarini et al., 2018 [74]	BMC	E: 1046 (winter) NE: 1046 (summer)	0.42 ± 0.54	0.22 ± 0.34	1.91	*p* < 0.001	19
de Carvalho Cavalcante et al., 2017 [79]	BMC	E: 19 NE: 24	1.43 ± 1.0	0.12 ± 0.78	11.92	*p* < 0.05	12
Coronas et al., 2016 [80]	PBL	E: 41 NE: 17	0.93 ± 0.09	0.41 ± 0.08	2.27	*p* < 0.001	11
	BMC	E: 38 NE: 19	0.14 ± 0.04	0.16 ± 0.05	0.875	*p* > 0.05	11
Mørck et al., 2016 [75]	PBL	E: 52 NE: 48	2.21 ± 1.5	1.89 ± 1.5	1.17	*p* > 0.05	16
de Silveira Fleck et al., 2014 [81]	BMC	E: 33 NE: 34	4.57 ± 2.05	2.31 ± 1.10	1.98	*p* < 0.001	15
Ceretti et al., 2014 [76]	BMC	E: 97 NE: 25	3.1 ± 1.4	2.5 ± 0.8	1.24	n.a.	13
Demircigil et al., 2013 [84]	BMC	E: 93 (winter) NE: 93 (summer)	1.87 ± 1.66	2.73 ± 1.98	0.68	*p* = 0.001	20
Sisenando et al., 2012 [82]	BMC	E: 245 NE: 128	1.43 ± 0.84	0.29 ± 0.41	4.93	*p* < 0.01	14
Pedersen et al., 2006 [77]	WB	E: 23 NE: 24	8.0 ± 3.3	5.8 ± 3.4	1.38	*p* > 0.05	16
Lahiri et al., 2000 [85]	BMC	E: 153 NE: 116	2.2 ± 0.4	1.7 ± 0.3	1.29	*p* < 0.05	13

^[a]^ BMC, buccal mucosa cells; PBL, peripheral blood lymphocytes; WB, whole blood/leukocytes. ^[b]^ n.a., not available. ^[c]^ Quality score.

## Data Availability

All the data supporting the reported results of this meta-analysis are included within the article or in the enclosed Appendix A.

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
