# Peer review of "Cytogenetic Effects in Children Exposed to Air Pollutants: A Systematic Review and Meta-Analysis"

_ijerph, 2022, doi:10.3390/ijerph19116736_

Round 1
Reviewer 1 Report
This article explored the relationship between the exposure to environmental pollutants and micronucleus frequency in child groups, and the research is interesting. .However, there are still some limits need to be revised.
- The results in abstract are rarely described, particularly not efficiently described the characteristics of the included literature
- The background part is lengthy and wordy. I suggest authors to briefly introduce the concept of air pollution, harm (especially for children), cytogenetic significance and the purpose of this article.
- The publication bias is not specified in the results section. The funnel diagram is too concentrated, which can be revised, and add relevant explanations to the results section.
- The discussion part was not complete, which need to be revised. After summarizing the main findings of this study, there was no discussion on the publication bias, literature quality or other influencing factors that affected the conclusions of the study.
- After explaining the research advantages and limitations, the implications or practical significance of the research findings for future research are not discussed.
- Only three databases were searched in the literature, Cochrane and Embase, two important databases, were not searched and less literature was included.
Author Response
Please, see attached file.

Reviewer 2 Report
Congratulations on an excellent well-presented article.
1: A few minor English corrections
Pg1, Line 28 delete Actually
Pg1, Line 36 add such as PM [formed from secondary organic aerosols] and ..
Pg1, Line 41 ultrafine particles are a relatively new "grouping". Please add a reference.
Pg 2, Line 78, delete "As said,"
Pg2, Line 89 rephrase, perhaps "PM is the most studied air pollutant of health concern"?
Pg 4, Line 176, Excel or MS Word?
2: Study design, Table 1 - Country mostly Italy (3) or Brazil (5) compared to other (6). I suspect your search criteria inadvertently excluded other regions such as the USA. Nevertheless, I'm satisfied that your sample population size was acceptable.
Author Response
Please, see attached file,
